# The Cytosolic Acetoacetyl-CoA Thiolase TaAACT1 Is Required for Defense against *Fusarium pseudograminearum* in Wheat

**DOI:** 10.3390/ijms24076165

**Published:** 2023-03-24

**Authors:** Feng Xiong, Xiuliang Zhu, Changsha Luo, Zhixiang Liu, Zengyan Zhang

**Affiliations:** 1Hunan Provincial Key Laboratory of Forestry Biotechnology, College of Life Science and Technology, Central South University of Forestry and Technology, Changsha 410004, China; 2The National Key Facility for Crop Gene Resources and Genetic Improvement, Institute of Crop Sciences, Chinese Academy of Agricultural Sciences, Beijing 100081, China

**Keywords:** cytosolic acetoacetyl-CoA thiolase, *Fusarium pseudograminearum*, jasmonate, plant defense, wheat (*Triticum aestivum*)

## Abstract

*Fusarium pseudograminearum* is a major pathogen for the destructive disease Fusarium crown rot (FCR) of wheat (*Triticum aestivum*). The cytosolic Acetoacetyl-CoA thiolase II (AACT) is the first catalytic enzyme in the mevalonate pathway that biosynthesizes isoprenoids in plants. However, there has been no investigation of wheat cytosolic AACT genes in defense against pathogens including *Fusarium pseudograminearum*. Herein, we identified a cytosolic AACT-encoding gene from wheat, named *TaAACT1*, and demonstrated its positively regulatory role in the wheat defense response to *F. pseudograminearum*. One haplotype of *TaAACT1* in analyzed wheat genotypes was associated with wheat resistance to FCR. The *TaAACT1* transcript level was elevated after *F. pseudograminearum* infection, and was higher in FCR-resistant wheat genotypes than in susceptible wheat genotypes. Functional analysis indicated that knock down of *TaAACT1* impaired resistance against *F. pseudograminearum* and reduced the expression of downstream defense genes in wheat. TaAACT1 protein was verified to localize in the cytosol of wheat cells. *TaAACT1* and its modulated defense genes were rapidly responsive to exogenous jasmonate treatment. Collectively, TaAACT1 contributes to resistance to *F. pseudograminearum* through upregulating the expression of defense genes in wheat. This study sheds new light on the molecular mechanisms underlying wheat defense against FCR.

## 1. Introduction

The acetoacetyl-CoA thiolase (EC 2.3.1.9), also called thiolase II or Acetyl-CoA acetyltransferase, can catalyze the production of acetoacetyl-CoA from two molecules of acetyl-CoA [1,2]. It is highly conserved in eukaryotes and prokaryotes [3]. In eukarya, including plant species, the cytosol Acetoacetyl-CoA thiolase (AACT)/thiolase II is the first catalytic enzyme in the cytosolic mevalonate pathway required for the biosynthesis of isoprenoids [1,2,4,5,6,7]. Isoprenoids are a large, diverse class of hydrocarbons [5,8,9], and play roles in diverse cellular processes in eukaryotes [1,2,10,11]. For example, some isoprenoids function as defense compounds, are involved in phytohormone biosynthesis, and have an antioxidant function under abiotic and biotic stresses in *planta* [5,12,13,14,15,16]. However, the genetic study of cytosolic thiolases II is rare, since *Arabidopsis thaliana AACT* homozygous mutant (T-DNA mutant) is unviable [17] and the stable transgenic plants overexpressing thiolases II are very difficult to obtain [18]. A recent paper reported that the transgenic alfalfa (*Medicago sativa* L.) plants overexpressing *MsAACT1* (thiolase II) increased production of squalene and displayed enhanced tolerance to salt/cold stresses, suggesting that MsAACT1 positively participates in salt/cold stress adaptation by regulating the mevalonate pathway and isoprenoid biosynthesis [1]. However, no genetic study on the function of plant cytosolic AACTs, including wheat cytosolic AACT1, in defense against phytopathogens, has been reported yet.

Wheat (*Triticum aestivum* L.) is one of the important food sources all over the world. The soil-borne fungus *Fusarium pseudograminearum* is a main pathogen for Fusarium crown rot (FCR) disease of cereal crops, including wheat. FCR has been become one of the most destructive diseases, seriously threatening the production of wheat in many regions of the world [19,20,21,22]. For example, in Australia and the Pacific Northwest region of the United States, FCR can reduce wheat yield by 35% in severely affected fields [19,20]. Recently, the severity and incidence of FCR has rapidly increased in major wheat-producing regions of China [22]. The planting of disease-resistant varieties is an efficient and environmentally friendly way to control FCR. No wheat germplasm with complete resistance to FCR has been identified, and only some partially resistant genotypes are available [23]. Many quantitative trait loci affecting FCR resistance have been identified [22,23,24]. A recent paper reported that the dirigent gene TaDIR-B1 was a negative contributor to FCR resistance in wheat [25]. The wall-associated kinase TaWAK-6D and DUF26 domain-containing receptor-like kinase TaCRK-7A have been shown to be required for defense responses to *Fusarium pseudograminearum* infection in wheat [26,27]. However, little is known about involvement of cytosolic AACT/thiolase II in wheat in innate immune responses to *F. pseudograminearum* infection.

In this study, in order to explore whether and how cytosolic AACT/thiolase II plays a role in wheat defense, we identified an Acetoacetyl-CoA thiolase-encoding gene in wheat, *TaAACT1*, through haplotype and transcriptional analyses. Subsequently, we analyzed the transcript profile of *TaAACT1* in wheat, and investigated the function in the innate immune response of wheat to *Fusarium graminearum* infection by a combination of VIGS technology and disease assessment. In addition, the cytosolic localization of TaAACT1 protein was validated. This is the first genetic investigation about the function of a cytosolic AACT/thiolase II-encoding gene in plant immunity.

## 2. Results

### 2.1. Identification of TaAACT1 Gene Involved in Wheat Resistance to FCR

By means of bulked segregant RNA sequencing (BSR-Seq) [28] toward the recombinant inbred lines (RILs derived from CI12633/Yangmai 9, F_16_) combined with wheat variant omics (WheatUnion, http://wheat.cau.edu.cn/WheatUnion/), we analyzed single-nucleotide polymorphisms (SNPs) among the *TaAACT1* gene (TraesCS3D02G005500.1) in FCR-resistant wheat cultivars [CI12633, Chinese spring (CS), Jinan 13, Nonglin 10, and Mianyang 26] and its allelic sequences in susceptible wheat cultivars (Handan 6127, Jinan 17, Yangmai 9, and Yangmai158). As a result, 50 SNP sites were found among *TaAACT1* gene sequences of the different wheat cultivars, and the majority (47 SNPs) distributed among the last 1300 bp sequence of *TaAACT1* (nucleotide no. 2028th to 3327th). Interestingly, 13 nucleotide allelic variants seemed to be associated with FCR resistance/susceptibility of wheat and formed at least 2 haplotypes (Table 1, Figure 1), of which haplotype I was specific to the tested FCR-resistant wheat cultivars (CI12633, CS, Jinan 13, Nonglin 10, and Mianyang 26) but haplotype II was present in the tested susceptible wheat cultivars (Handan 6127, Jinan 17, Yangmai 9, and Yangmai158). As shown in Table 1 and Figure 1, of the 13 SNPs related to FCR resistance, 10 are located in the intron regions of this gene, and the remaining 3 (3156th, 3304th, and 3315th) are located in the gene-coding region. Among the three SNP sites in the coding region, two single-nucleotide mutations are synonymous mutations, which do not cause changes in the amino acid sequence, and one single-nucleotide mutation is a nonsynonymous mutation, which changes the encoded amino acid from serine (S) in the resistant wheat cultivars to proline (P) in the susceptible wheat cultivars.

Reverse-transcription quantitative PCR (RT-qPCR) analysis showed that toward *F. pseudograminearum* infection, the transcript level of *TaAACT1* in stems of FCR-resistant wheat cultivar CI12633 was raised, reaching its peak at 4 days post inoculation (dpi) (about 3-fold over non treatment) (Figure 2A). Subsequently, we examined the gene transcript induction in the tissues of the resistant wheat cultivar CI12633, and the result showed that compared with the nontreatment, the expression level of *TaAACT1* in all the tested tissues was elevated after 4 dpi with *F. pseudograminearum*, and the highest induction occurred in the wheat stems (about 4.8-fold over noninoculation), then in the wheat roots (about 3.9-fold over noninoculation) of the wheat cultivar CI12633 (Figure 2B). In fact, the FCR disease symptom majorly appears in stems and roots of wheat plants [19,20,22]. Furthermore, we tested the transcript level of *TaAACT1* in the stems of five different wheat cultivars with different FCR disease indexes [26,27] at 4 dpi with *F. pseudograminearum*. Expectedly, the transcript level of *TaAACT1* was higher in FCR-resistant wheat cultivars than susceptible wheat cultivars (Figure 2C). These results suggest that change in expression of *TaAACT1* was associated with wheat resistance to *F. pseudograminearum* infection, and TaAACT1 might be involved in wheat defense against *F. pseudograminearum* infection.

### 2.2. Sequence and Phylogenetic Characteristics of TaAACT1

The full-length coding sequence and genomic sequence of *TaAACT1* were cloned from the partially resistant wheat cultivar CI12633 and had 100% identity with those of CS (TraesCS3D02G005500.1). The *TaAACT1* gene contains an open reading frame (ORF) with 1257 bp length. Comparison of the genomic and cDNA sequences showed that the genomic sequence of *TaAACT1* was comprised of twelve introns and thirteen exons (Figure 1A). The predicted protein TaAACT1 consists of 418 amino acid (aa) residues with a predicted Mw of 43.02 kDa and theoretical pI of 8.41. TaAACT1 protein has a Thiolase-N domain and a Thiolase-C domain (Figure 3A,B).

Thiolases in plant cells are divided into biosynthetic thiolases and degradative thiolases according to their functions. Degradative thiolases, also known as 3-ketoacyl-CoA thiolase (KAT*)*, belong to peroxisomal thiolases I and can catalyze β-oxidation of fatty acid. Biosynthetic thiolases, also known as Acetoacetyl-CoA thiolases (AACT), belong to cytosolic thiolase II enzymes and catalyze the condensation of two acetyl-CoA to form acetoacetyl-CoA [1,4,11]. To determine the phylogenetic relationship of TaAACT1, we constructed a phylogenetic tree containing TaAACT1 and 15 other thiolase II (AACT/ACAT) proteins as well as 3 thiolase I proteins of other species (Figure 2D). The analysis of phylogenetic tree showed that thiolase I and thiolase II proteins were categorized into two groups (Figure 3C). In the thiolase II group, TaAACT1 is clustered with *Aegilops tauschii* AACT (XP_0201631811), *Triticum dicoccoides* AACT (XP_0374108261), *Lolium rigidum* AACT (XP_0470549061), rice (*Oryza sativa*) AACT (BAB398721), and corn (*Zea mays*) AACT (ACG347351) into one branch, while the cytosolic thiolase II (ACAT/AACT) proteins of dicots, namely *Arabidopsis thaliana* ACAT2 (BAH19918), *Medicago sativa* AACT1(ACX474701), radish (*Raphanus sativus*) AACT (CAA550061), and tobacco (*Nicotiana tabacum*) AACT (AAU956181), were clustered into another branch. AtKAT1 (AEE277361) and AtKAT2 (BAA252481) of *Arabidopsis thaliana* and rice thiolase I (AAO725881) were categorized to the peroxisomal thiolase I group. Additionally, TaAACT1 is closely related to the corresponding protein of *Aegilops tauschii* with 100.00% identity at the full-length protein level. These results indicate that TaAACT1 belongs to the cytosolic thiolase II group and that TaAACT1 in wheat might originate from that of *Aegilops tauschii*.

### 2.3. TaAACT1 Is Required for Wheat Resistance to F. pseudograminearum Infection

The virus-induced gene silencing (VIGS) mediated by barley stripe mosaic virus (BSMV) was used to investigate the defense function of *TaAACT1* against *F. pseudograminearum* infection. The gene-silenced fragment is predicted through Si-Fi software (Figure 4A). The optimum fragment with 194 bp length (covering 166 bp ORF and 28 bp 3’-UTR) of *TaAACT1* was subcloned in an antisense orientation into the multicone site of BSMV RNAγ, resulting in the BSMV:*TaAACT1* construct (Figure 4B). When the BSMV:GFP/BSMV:*TaAACT1* RNAs were transfected to the newly emerging leaves of the mildly-resistant wheat cultivar CI12633 plants at the 3-leaf-stage for 14 days, the symptoms of BSMV infection appeared on the newly emerging 4th leaves and the transcript of the BSMV coat protein gene could be detected in the transfection wheat leaves (Figure 5A). These indicated that the BSMV successfully infected these wheat plants. RT-qPCR analysis showed that compared with the BSMV:*GFP*-infected CI12633 wheat plants (control), the transcript level of *TaAACT1* in BSMV:*TaAACT1*-infected CI12633 was significantly decreased (Figure 5B), suggesting that the *TaAACT1* gene was successfully silenced in these wheat plants mediated by BSMV.

To assess the defensive role of *TaAACT1* against *F. pseudograminearum* infection, *TaAACT1*-silenced and BSMV:*GFP*-infected wheat plants were inoculated with *F. pseudograminearum* WHF220 at the same time. During a period of 28 dpi with *F. pseudograminearum*, compared with the BSMV:*GFP*-infected wheat plants, the stems of *TaAACT1*-silenced CI12633 wheat plants displayed larger lesion sizes of FCR (Figure 5C). In the two experimental batches, the average lesion length on the stems of *TaAACT1*-silenced CI12633 plants was 4.10 cm, whereas that of BSMV:*GFP*-infected CI12633 plants was 2.43 cm; the average lesion width on the stems of *TaAACT1*-silenced CI12633 plants was 0.74 cm, whereas that of BSMV:*GFP*-infected CI12633 plants was 0.35 cm (Figure 5D). The disease tests showed that the average values of infection types (ITs) on the stems of *TaAACT1*-silenced CI12633 plants were 4.57 and 4.15, respectively, while the average values of ITs on the stems of BSMV:*GFP*-infected CI12633 plants were 2.59 and 2.03, respectively (Figure 5E). The above data suggest that the expression of *TaAACT1* is required for wheat resistance to *F. pseudograminearum* infection. 

### 2.4. TaAACT1 Positively Regulates the Expression of Defense Genes

Plant chitinases and defensin have shown to function in directly restricting the growth of pathogenic fungi [29,30,31,32,33,34,35]. In order to explore if *TaAACT1* regulates expression of these defense genes during the wheat immune response against *F. pseudograminearum* infection, we tested the transcript levels of wheat *TaChitinase 2* (*TaChit2*), *TaChitinase 3* (*TaChit3*), *TaChitinase 4* (*TaChit4*), and *TaDefensin* through RT-qPCR in the *TaAACT1*-silenced wheat plants and the BSMV:*GFP*-infected wheat plants inoculated with *F. pseudograminearum* for 4 d. The results show that the transcript levels of *TaChit2*, *TaChit3*, *TaChit4*, and *TaDefensin* were significantly decreased in the more susceptibly *TaAACT1*-silenced wheat plants compared to the BSMV:*GFP*-infected wheat plants (Figure 6). These results suggest that *TaAACT1,* acting as an upstream regulator, positively modulates the expression of *TaChit2*, *TaChit3*, *TaChit4*, and *TaDefensin* during wheat defense responses to *F. pseudograminearum* infection.

### 2.5. TaAACT1 and Its Modulated Defense Genes Are Responsive to an Exogenous Jasmonic Acid Stimulus

The promoter sequence of *TaAACT1* contains two jasmonic acid (JA)-responsive *cis*-acting elements (CGTCA motif/TGACG motif) and two abscisic acid (ABA)-responsive *cis*-acting elements (ABRE, ACGTG motif/GCCGCGTGGC motif). To explore whether *TaAACT1* is responsive to JA and ABA, we tested the transcript level of *TaAACT1* in the wheat cultivar CI12633 seedlings sprayed with exogenous MeJA (0.05 mM) and ABA (0.1 mM). As results, upon exogenous MeJA treatment, the expression level of *TaAACT1* was significantly increased from 0.5 to 3 h, and reached its peak (~3.40-fold compared with the mock (H_2_O treatment)) at 0.5 h (Figure 7A). However, exogenous ABA treatment did not significantly affect the expression of *TaAACT1* (Figure 7B). Moreover, we also examined the transcript profiles of the *TaAACT1*-regulated defense genes *TaChit2*, *TaChit3*, *TaChit4,* and *Defensin* in the wheat cultivar CI12633 leaves treated with exogenous MeJA for 0.5 h and 3 h. The results showed that compared with the mock (H_2_O-treatment), the transcript levels of *TaChit2*, *TaChit3*, *TaChit4*, and *Defensin* were significantly and rapidly increased by exogenous MeJA (Figure 7C). Taken together, the data suggest that the expression of *TaAACT1* and its regulated defense genes (*TaChit2*, *TaChit3*, and *Defensin*) were all in response to the stimulation of exogenous MeJA, and that *TaAACT1*, *TaChit2*, *TaChit3*, *TaChit4* and *TaDefensin* should be located in the JA signaling pathway.

### 2.6. TaAACT1 Protein Localizes in the Cytosol in Wheat

The subcellular localization of TaAACT1 protein was predicted by PSORT Ⅱ online software. The predicted results are as follows: thiolase II TaAACT1 mainly exists in the cytoplasm, and a small part in the chloroplast. In order to determine the subcellular location of TaAACT1, the full coding sequence of *TaAACT1* without the stop codon was subcloned in fusion to the N-terminus of green fluorescent protein (*GFP)-coding gene*, generating the p35S::*TaAACT1-GFP* fusion-expressing vector. The resulting p35S::*TaAACT1-GFP* and the control p35S::*GFP* were separately introduced into wheat mesophyll protoplasts with the PEG method and transiently expressed in the wheat cells. These fluorescent proteins expressed were observed via a confocal microscope. The results showed that fluorescence images of 35S::TaAACT1-GFP were distributed in the cytosol, while fluorescence images of the GFP control protein were distributed in both the cytoplasm and nucleus (Figure 8). The experimental results suggest that TaAACT1 indeed localizes in the cytosol of wheat, verifying the above prediction for cytoplasm. 

## 3. Discussion

Wheat is an important staple crop and provides 20% of the food calories consumed by human beings worldwide [36]. Its efficient production is often threatened by abiotic and biotic stresses, such as FCR disease. The planting of wheat with resistance is one optimal strategy to control disease. Recently, the dirigent gene *TaDIR-B1* (a negative contributor) and two positive regulators, namely TaWAK-6D and TaCRK-7A, were identified to be involved in resistance responses of FCR in wheat [25,26,27]. However, there have been limited papers about FCR-resistant genes and molecular mechanisms underlying wheat defense against *Fusarium pseudograminearum* infection. Additionally, a recent study showed that the overexpression of alfalfa thiolase II *MsAACT1* in transgenic plants enhanced salinity tolerance and the production of squalene in salt-stress conditions [1]. However, little is known about role of cytosolic thiolase II /AACT1 in the plant’s innate immunity to the infection of pathogens.

In the current research, we identified the wheat cytosol thiolase II-encoding gene *TaAACT1* associated with FCR resistance of wheat, and provided evidence that *TaAACT1*, acting as a positive contributor, was required for host defense against *F. pseudograminearum* infection. By means of analyses in BSR-seq and wheat variant omics, 13 SNPs in *TaAACT1*, related to FCR resistance, were identified. It is very interesting to develop functional markers based on these SNPs for molecularly breeding wheat with resistance to FCR in future. Transcript analysis results showed that the transcript induction of *TaAACT1* by infection of *F. pseudograminearum* was higher in FCR-resistant wheat cultivars than in susceptible wheat cultivars, and the increase occurred in stems/roots at the greatest extent. Similarly, the expression levels of *AtACAT2* and *MsAACT1* were higher in roots than in leaves in *Arabidopsis thaliana* and alfalfa under optimal growth conditions [1,6]. Functional studies on plant thiolases by the genetic method have been rare, since the overexpression of these enzymes during the transformation/regeneration procedure is sufficient to avoid the generation of stable transgenic plants [18], and thiolases II from dicots have an extremely high nucleotide identity, making the RNAi strategy difficult to design with the certainty of ensuring its specificity for plant species [1], and *Arabidopsis* mutant of *AACT 2* with only one copy is nonviable [17]. Fortunately, the VIGS technique can be used as an alternative approach to rapidly analyze gene function [37,38]. For example, due to a lack of generation of transgenic wheat overexpressing a wheat cytochrome P450 gene *TaCYP72A*, VIGS was used to conduct functional analyses, and the results showed that *TaCYP72A* enhances resistance to deoxynivalenol and grain number [38]. Here, the VIGS technique was used to study the defensive function of *TaAACT1*, and the results show that the silencing of *TaAACT1* decreased resistance to FCR caused by infection of *F. pseudograminearum.* To our knowledge, this is the first investigation about the role of plant thiolases in plants’ innate immunity. This study deepens the understanding of plant defense mechanisms against pathogens.

In plant species, many chitinases and defensin peptides show antifungal activity [29,30,35,39]. Chitinases can hydrolyze the chitin (the major structural component) of fungal cell walls, in turn directly inhibiting the growth of pathogenic fungi [29,32]. Antifungal activity of plant defensin requires specific binding to membrane targets to inhibit the growth of fungi [30,31,39]. Additionally, chitinases also have the secondary effect of releasing chitin monomers, which acts as a pathogen/damage-associated molecular pattern for the activation of the plant immune system [40]. Many chitinases and defensin-encoding genes are induced in response to infection by pathogens [41,42], and are also upregulated by upstream immune genes in fungal-infected plants [26,27,43,44,45,46,47,48,49,50]. The current research indicates that expression of *TaAACT1* is required for the expression of *TaChitinase 2*, *TaChitinase 3*, *TaChitinase 4*, and *TaDefensin* in wheat inoculated by *F. pseudograminearum*, suggesting that *TaChitinase 2*, *TaChitinase 3*, *TaChitinase 4* and *TaDefensin* act downstream of *TaAACT1* and are positively modulated by upstream *TaAACT1*. JA is a primary phytohormone associated with plant resistance responses to necrotrophic pathogens [51]. Some kinases and transcription factors, as well as upstream proteins, can respond and transduce JA signaling to the downstream genes [46,47,52]. For instance, the expression of TaSTT3b-2B was significantly induced from 1 to 8 h after MeJA treatment; overexpression of *TaSTT3b-2B* in transgenic wheat increased endogenous JA content and the expression of JA synthesis-related genes and enhanced resistance to sharp eyespot and grain weight [46]. However, little is known about the effect of JA on thiolase II in wheat plants. Our analyses showed that *TaAACT1* rapidly responded to exogenous MeJA stimulus, and MeJA treatment also significantly induced the expression of *TaAACT1*-modulated defense genes (*Chitinases 2/3/4* and *defensin*). Thus, JA signaling might play an important role in *TaAATC1*-mediated immune responses to *F. pseudograminearum* infection. Taken together, these results suggest that TaAACT1 positively contributes to the resistance of wheat against *F. pseudograminearum* infection by regulating the expression of *TaChitinase 2*, *TaChitinase 3*, *TaChitinase 4*, and *TaDefensin* in JA signaling. 

Protein sequence analysis showed that the TaAACT1-predicted protein contains two domains: a Thiolase-N domain and a Thiolase-C domain, similar to those of *Arabidopsis thaliana* cytosolic Thiolase II (AtACAT2, BAH19918), and alfalfa cytosolic Thiolase II/AACT1 (MsAACT1, ACX474701), involved in the mevalonate pathway [1,6,7]. Our phylogenetic analysis indicated that the TaAACT1 protein, together with *Arabidopsis thaliana* cytosolic Thiolase II/AtACAT2 and MsAACT1, as well as other cytosolic Thiolase II proteins, are classified into the same group, and that TaAACT1 in wheat might originate from that of *Aegilops tauschii*. The subcellular localization experimental assay indicated that TaAACT1 protein indeed localizes in the cytosol of wheat, like the AtACAT2 subcellular localization result [6,7]. In a recent paper, the alfalfa cytosolic Thiolase II MsAACT1 was determined to possess thiolase activity, not only using bacteria as a heterologous system but also in *planta*, and that the expression of MsAACT1 increased abiotic stress resistance in bacteria and in *planta*, strongly suggesting that MsAACT1 positively regulates abiotic stress tolerance by increasing squalene biosynthesis [1]. Additionally, infection with avirulent pathogens, *tobacco mosaic virus*, or *Pseudomonas syringae* pv. tabaci induced accumulation of polyisoprenoid alcohols in the leaves of resistant tobacco plants but did not cause an increase in polyisoprenoid content in susceptible tobacco plants [15]. In the furture, it is very interesting to investigate if TaAACT1 is required for isoprenoid biosynthesis during wheat resistance responses to *F. pseudograminearum.*

## 4. Materials and Methods

### 4.1. Plant and Fungal Materials, Primers, and Treatments

Five wheat cultivars, including *FCR*-resistant cultivars (Nivat14, CI12633, Chinese Spring), and susceptible cultivars (Yangmai 6 and Yangmai 158), were used to examine transcript profiles of *TaAACT1* with the *F. pseudograminearum* inoculation. 

All of the wheat plants were grown in a 16 h light/8 h dark (22 °C/12 °C) regimen. At the tillering stage, small toothpick fragments harboring the well-developed mycelia of *F. pseudograminearum* were inoculated between the second base sheaths and stems of wheat plants [26]. The inoculated sheaths of the resistant wheat cultivar CI12633 were collected at noninoculation or 1, 2, 3, 4, 5, 6, or 7 dpi with *F. pseudograminearum*. The inoculated sheaths of Nivat 14, Chinese spring, CI12633, Yangmai 9, and Yangmai 158 were separately sampled at 4 dpi with *F. pseudograminearum* for RNA extraction. These samples were quickly frozen in liquid nitrogen and stored at −80 °C prior to extraction of total RNA. The seedlings of the wheat cultivar CI12633 at the two-leaf stage were separately treated with 0.05 mM MeJA and 0.1 mM ABA dissolved in ultrapure water, and ultrapure water solution (mock, as control) for noninoculation and 0.5, 1, 3, 6, and 12 h, as described by Zhang et al. [43]. The treated leaves were collected for RNA extraction.

The pathogenic fungus *F. pseudograminearum* strain WHF220 was isolated from the FCR-symptomatic wheat sheaths in Shandong by Prof. Jinfeng Yu and Dr. Li Zhang (Shandong Agricultural University, Tai’an, China).

All primers and their sequences in the study are shown in Appendix A.

### 4.2. RNA Extraction, cDNA Synthesis, and RT-qPCR Analysis

Total RNA was extracted from wheat samples tested using Trizol reagent (Invitrogen, Carlsbad, CA, USA) according to the manufacturer’s instruction. RNA was purified and then checked for integrity. Total RNA was reverse-transcribed to cDNA using the FastQuant R T Kit (Tiangen, Beijing, China); RT-qPCR and specific primers were employed to measure expression levels of *TaAACT1*, *TaChit2*, *TaChit3*, *TaChit4*, and *TaDefensin* in wheat. RT-qPCR was performed using a SYBR Premix Ex Taq kit (TaKaRa, Tokyo, Japan) in a volume of 20 µL on an ABI 7500 instrument (Applied Biosystems, Foster City, MA, USA). Reactions were set up with the following thermal cycling profile: 95 °C for 3 min followed by 40 cycles of 5 s at 95 °C, 15 s at 55 °C, and 32 s at 72 °C, and completed with a melting curve analysis program. All RT-qPCRs were repeated three times. The relative expression of the tested gene was calculated with the 2^−ΔΔCT^ method [53], where the wheat *Actin* gene was used to normalize amounts of cDNA among the samples. 

### 4.3. SNP Analysis

The purified RNAs were extracted from the resistant and susceptible RILs (CI12633/Yangmai 9, F_16_) at 20 dpi and used to BSR-seq using the Illumina HiSeq 2500 sequencing technology platform at Beijing Biomarker Technologies. For each pool, three biological replicates were performed. After discarding low-quality raw reads, the generated 6 Gb clean reads from each library were mapped to wheat genomes. RNA-seq data were analyzed as described previously by Wang et al. [28] The SNP results of RNA-Seq analysis were correlated with the wheat variant omics analysis toward different FCR-resistant/susceptible wheat cultivars. Taken together, the SNPs in *TaAACT1* related to FCR resistance of wheat were identified.

### 4.4. Sequence and Phylogenetic Analyses of TaAACT1

One pair of primers (TaAACT1-F1/TaAACT1-R1) for TaAACT1 were designed and used to amplify the full ORF sequence of TaAACT1 from the cDNA of the wheat cultivar CI12633.The predicted protein sequence was analyzed with the Compute pI/Mw tool (http://web.expasy.org/compute_pi/ (accessed on 25 July 2022)) to determine the theoretical pI (isoelectric point) and Mw (molecular weight), The conserved domains and motifs in the deduced protein sequence were predicted by SMART (http://smart.embl-heidelberg.de/, accessed on 25 July 2022). Sequences were aligned via DNAMAN software. Subcellular localization of TaAACT1 protein was predicted using PSORT Ⅱ online software (https://psort.hgc.jp/form2.html, accessed on 25 July 2022). A phylogenetic tree was constructed by MEGA 11.0 (https://www.megasoftware.net/, accessed on 25 July 2022) software. Cis-elements in the TaAACT1 promoter (2000 bp) were analyzed using the PlantCARE (http://bioinformatics.psb.ugent.be/webtools/plantcare/html/, accessed on 25 July 2022). The JA- and ABA-responsive cis-acting elements in promoter of TaAACT1 are shown in Appendix A.

### 4.5. VIGS of TaAACT1 in Wheat CI12633

A short fragment (194 bp) derived from the coding sequence of *TaAACT1* was subcloned in antisense orientation into the multicloning site of the γ chain of the BSMV vector. According to the protocols described previously [44,45,46,48], BSMV:*TaAACT1* and BSMV:*GFP* viruses were transfected into the leaves of the wheat cultivar CI12633 in order to knock down endogenous *TaAACT1* transcript in the wheat plants. After 14 d, the fourth leaves of inoculated seedlings were harvested. These samples were quickly frozen in liquid nitrogen and stored at −80 °C prior to extraction of total RNA.

### 4.6. Assessment and Scoring of FCR Disease in BSMV-VIGS Wheat Plants 

About 21 d post-transfection with BSMV, the *TaAACT1*-silenced and BSMV:*GFP*-infected (control) wheat plants were inoculated with well-developed *F. pseudograminearum* mycelia toothpicks and kept at high humidity for 4 d in two batches. After 28 days of fungal inoculation, the average lesion length and width of FCR on the basal sheath of the above wheat plants were measured and recorded, and the FCR disease index was evaluated. The assessment methods of FCR disease severity were referred to the existing researches [22].

### 4.7. Subcellular Localization of TaAACT1 Protein

The coding region of TaAACT1 lacking the stop codon was amplified and added the restriction enzyme *Bam*H I site using the specific primers (TaAACT1-GFP-F/TaAACT1-GFP-R). The amplified fragment was digested with restriction enzyme *Bam*H I and subcloned in-frame into the N-terminus of the GFP coding region in the p35S::GFP vector, resulting in the TaAACT1-GFP fusion construct p35S::TaAACT1-GFP. The p35S::TaAACT1-GFP fusion construct and p35S::GFP control construct were separately individually introduced into wheat mesophyll protoplasts [54]. After incubation at 25 ℃ for 16 h, GFP signals were observed and photographed using a confocal laser scanning microscope (Zeiss LSM700, Jena, Germany) with a Fluor × 10/0.50 M27 objective lens and SP640 filter.

## 5. Conclusions

The wheat cytosol thiolase II-encoding gene *TaAACT1* was identified to be associated with FCR resistance of wheat. One haplotype of*TaAACT1* gene sequences in analyzed wheat genotypes was related to wheat resistance against FCR, and the gene transcript induction by *Fusarium pseudograminearum* infection was higher in FCR-resistant wheat cultivars than in susceptible wheat cultivars. Moreover, the transcript induction of this gene was the highest in stems then roots, where FCR disease primarily occurs. VIGS-based functional analyses indicated that the expression of *TaAACT1* was required for host defense responses against *F. pseudograminearum* infection, and *TaAACT1* positively modulated the expression of the defense genes *TaChitinase 2*, *TaChitinase 3 TaChitinase 4*, and *TaDefensin* in wheat. This study extends knowledge about wheat-*F. pseudograminearum* interactions and deepens our understanding of the molecular mechanisms underlying wheat plant immunity to fungal pathogens, especially *F. pseudograminearum*. The *TaAACT1* is a candidate gene for highly efficiently improving the resistance of wheat to FCR disease.

## Figures and Tables

**Figure 1 ijms-24-06165-f001:**
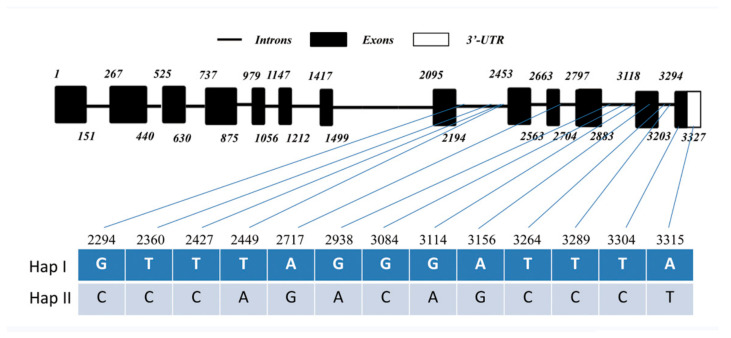
Genomic structure of the *TaAACT1* gene. Black squares indicate exons. Variations in the genomic sequence of TaAACT1 in the test wheat cultivars could be divided into two haplotypes (Hap I and Hap II). The number indicates the nucleotide site.

**Figure 2 ijms-24-06165-f002:**
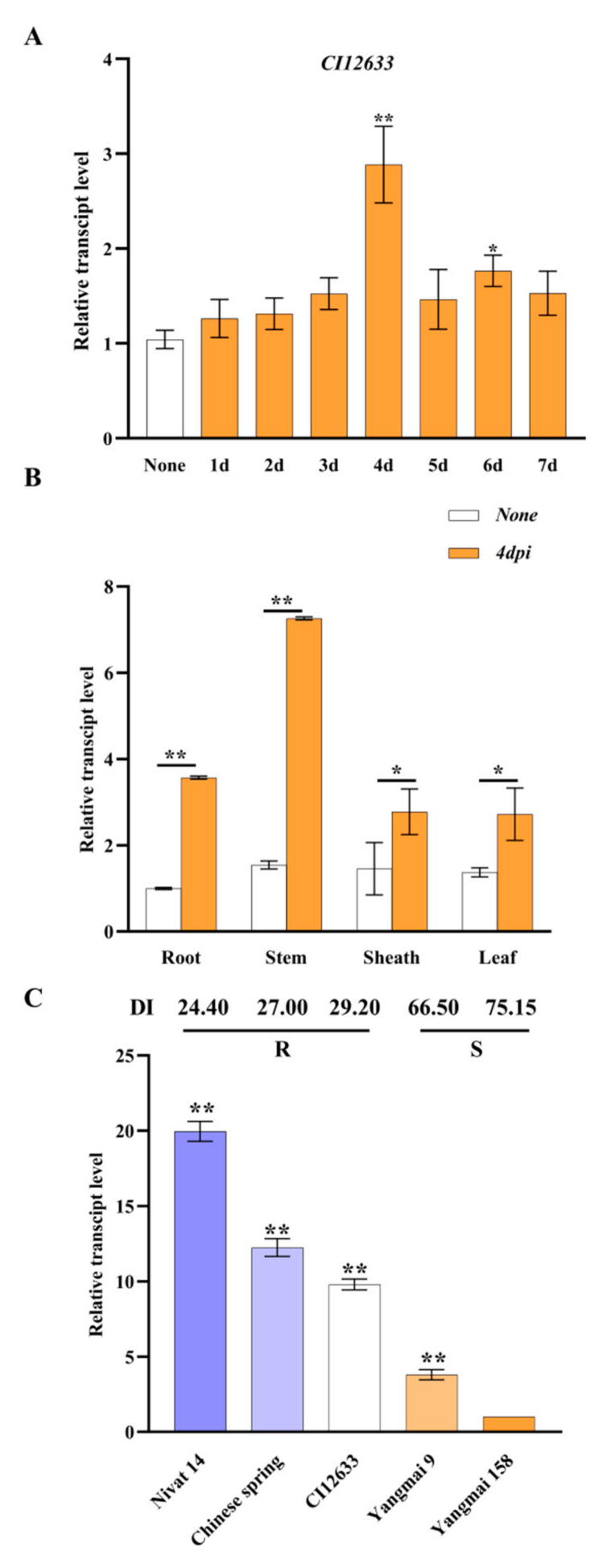
Spatiotemporal transcript patterns of *TaAACT1* in wheat toward infection of *F. pseudograminearum* strain WHF220. (**A**) Transcript abundance of *TaAACT1* in the middle-resistant wheat cultivar CI12633 stems inoculated with *F. pseudograminearum* for 1, 2, 3, 4, 5, 6, and 7 d. The gene transcript level at noninoculation (none) is set to 1. (**B**) Transcript abundance of *TaAACT1* in roots, stems, leaves, and sheaths of wheat cultivar CI12633 plants. The transcript level of *TaAACT1* in uninoculated roots was set to 1. (**C**) Transcript abundance of *TaAACT1* in five wheat cultivars at 4 d postinoculation with *F. pseudograminearum*. The *TaAACT1* transcript level in highly susceptible wheat cultivar Yangmai 158 was set to 1. DI represents the FCR disease index. R represents resistance and mild resistance to FCR. S represents susceptibility to FCR. *TaActin* was used as an internal control gene. Statistically significant differences were derived from the results of three independent replications (*t*-test: ** *p* < 0.01; * *p* < 0.05).

**Figure 3 ijms-24-06165-f003:**
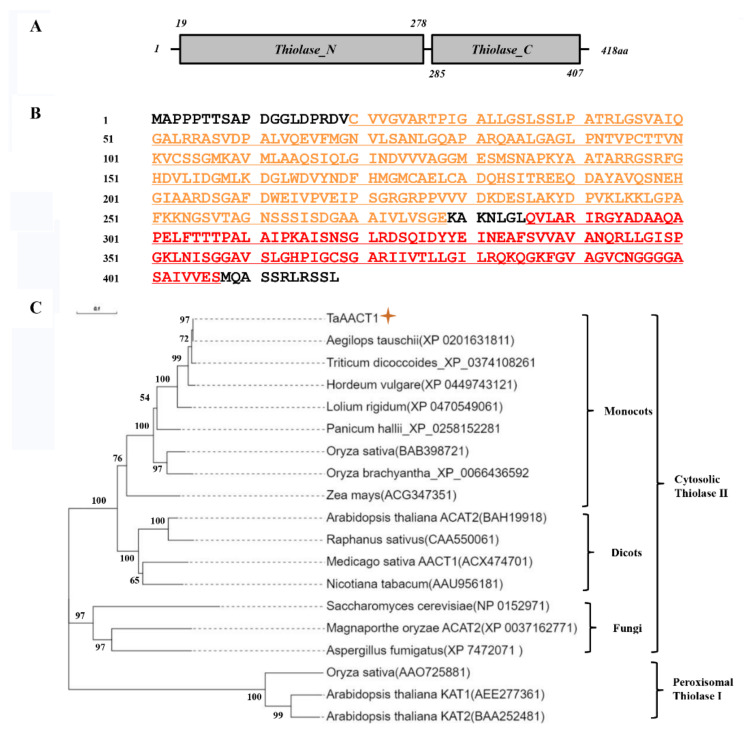
Sequence and phylogenetic analyses of TaAACT1 protein and 18 other thiolase proteins from other plant species and fungi. (**A**) Schematic diagram of two domains of TaAACT1 protein. (**B**) Sequence of TaAACT1 protein. The orange part indicates Thiolase-N domain. The red part indicates Thiolase-C domain. (**C**) The bootstrapped phylogenetic tree is constructed using the neighbor-joining phylogeny of MEGA 11.0. The scale represents the branch length, and each node reresents bootstrap values from 1000 replicates. † indicates the target protein; The bar (0.1) indicates 10% dissimilarity and distance scale can display the degree of difference.

**Figure 4 ijms-24-06165-f004:**
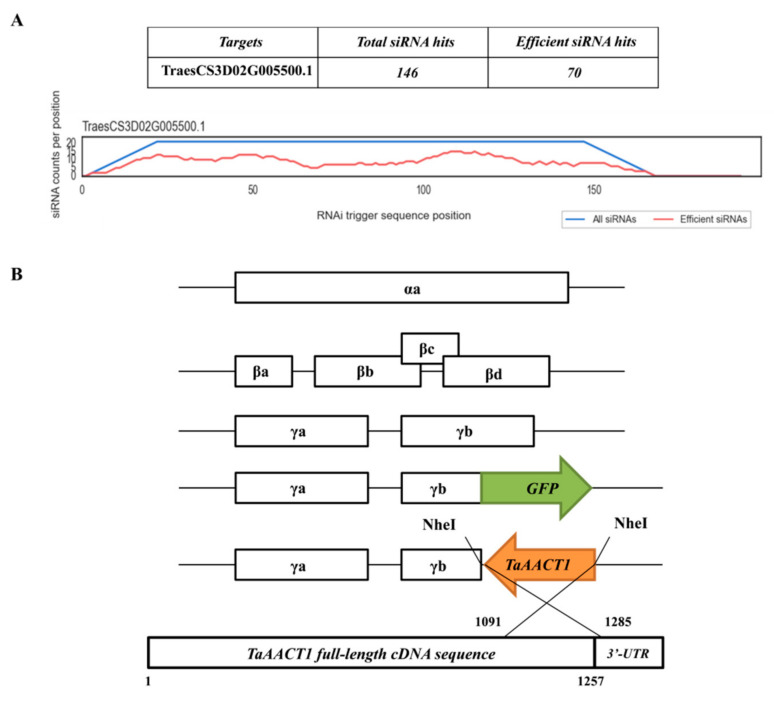
(**A**) Si-Fi software off-target prediction results. (**B**) Scheme of genomic RNAs of the BSMV and the recombinant virus construct BSMV: *TaAACT1*. The antisense orientation of the *TaAACT1* insert is indicated by orange box.

**Figure 5 ijms-24-06165-f005:**
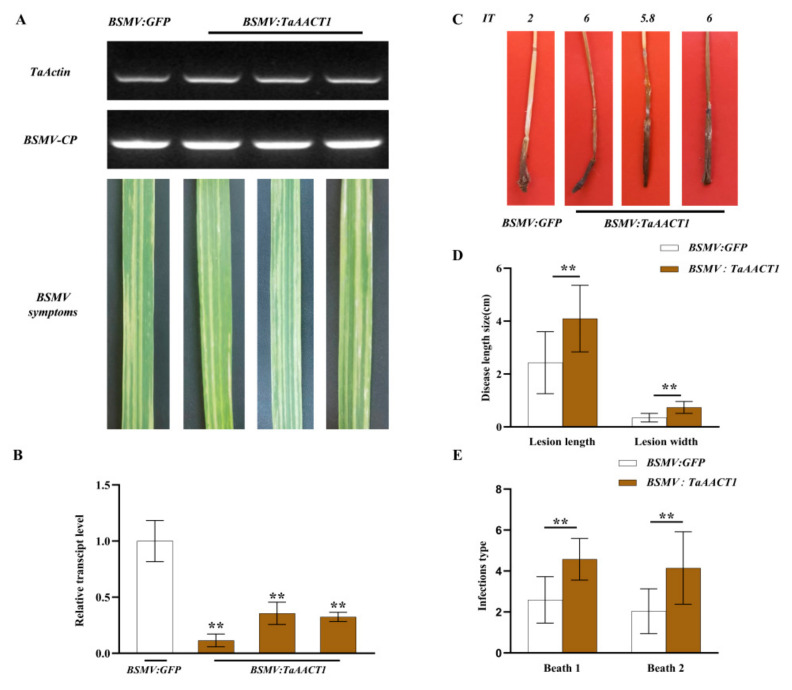
*TaAACT1* silencing compromises resistance of wheat CI12633 to *F. pseudograminearum*. (**A**) RT–PCR analysis of the transcript of BSMV coat protein (*CP*) in the wheat plants infected by BSMV:*GFP* or BSMV:*TaAACT1* for 14 d. BSMV-infected symptoms were observed on the wheat leaves at 14 d post-transfection with BSMV:*GFP* or BSMV:*TaAACT1*. (**B**) RT-qPCR analysis of the *TaAACT1* gene in the wheat plants infected by BSMV:*GFP* or BSMV:*TaAACT1* for 14 d. The relative transcript level of *TaAACT1* in BSMV:*GFP*-infected CI12633 plants was set to 1. *TaActin* was used as an internal control. (**C**) FCR symptoms on the stems of BSMV:*GFP*-infected and *TaAACT1*-silenced CI12633 plants at 28 d postinoculation (dpi) with *F. pseudograminearum* strain WHF220. (**D**) The average lesion sizes of the BSMV:*GFP*-infected and *TaAACT1*-silenced plants. (**E**) The infection types of the BSMV:*GFP*-infected and *TaAACT1*-silenced plants in two batches of disease tests ~28 d postinoculation (dpi) with *F. pseudograminearum*. ** (Student’s *t*-test, *p* < 0.01) represents statistically significant differences derived from at least three biological replications. Bars represent standard error of the mean.

**Figure 6 ijms-24-06165-f006:**
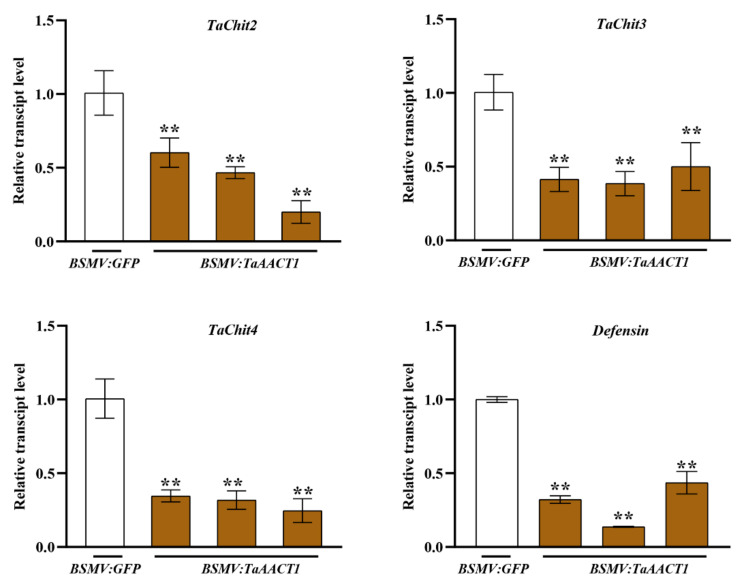
Transcript levels of *TaChitinase 2* (*TaChit2*), *TaChitinase 3* (*TaChit3*), *TaChitinase 4* (*TaChit4*), and *TaDefensin* in BSMV:*GFP*-infected and *TaAACT1*-silenced wheat CI12633 plants at 4 dpi with *F. pseudograminearum* WHF220. Relative transcript abundances of the tested genes (*TaChit2*, *TaChit3*, *TaChit4*, and *TaDefensin*) in *TaAACT1*-silenced CI12633 plants were quantified relative to thosein BSMV:*GFP*-infected CI12633 plants (set to 1). Statistically significant differences between *TaAACT1*-silenced and BSMV:*GFP*-infected CI12633 plants were determined based on three replications using Student’s *t*-test (*t*-test: ** *p* < 0.01). Bars represent standard error of the mean. *TaActin* was used as internal control.

**Figure 7 ijms-24-06165-f007:**
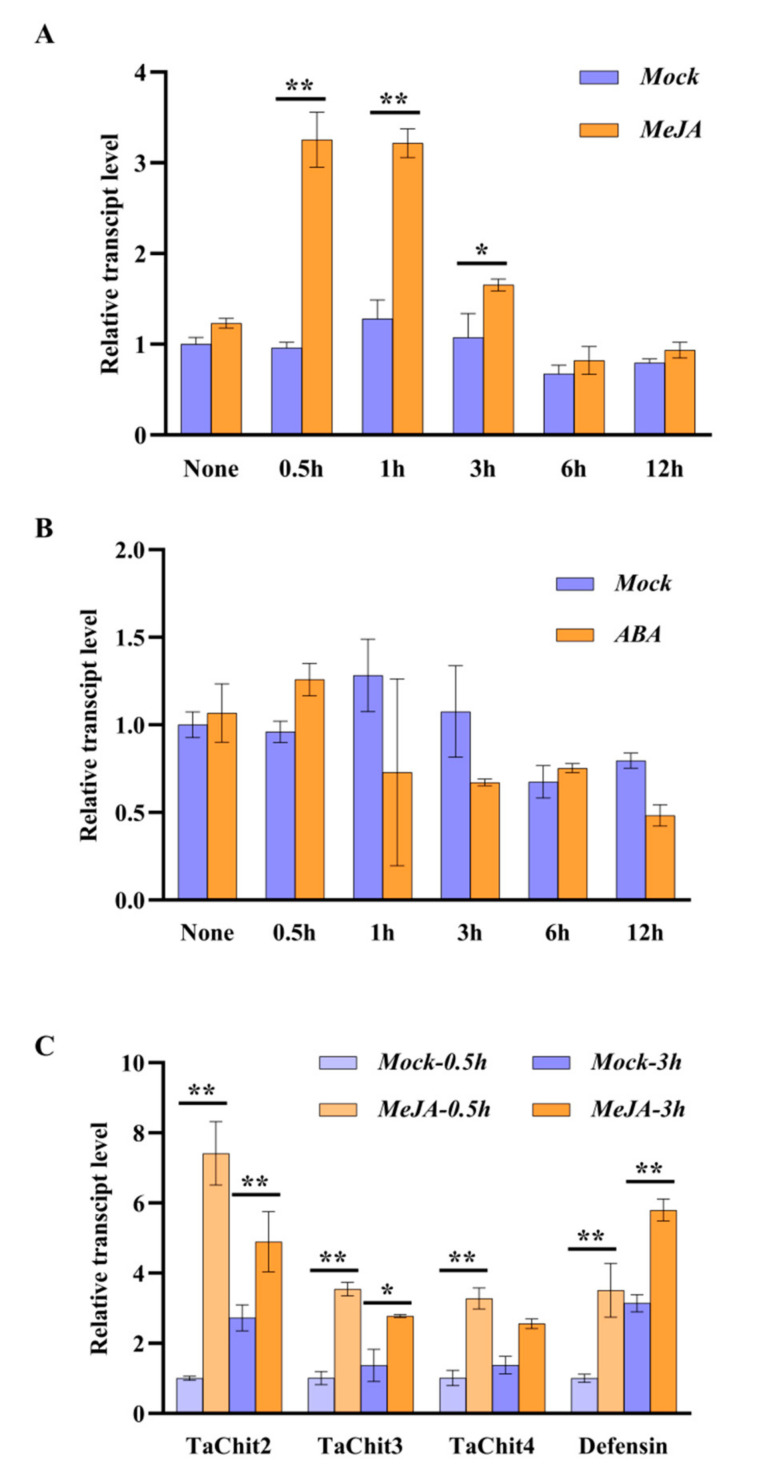
The transcript levels of *TaAACT1* and its regulated defense genes in wheat treated with exogenous MeJA and ABA. (**A**) Transcript level of *TaAACT1* in leaves of wheat cultivar CI12633 after exogenous application of 0.05 mM MeJA. The transcript level of *TaAACT1* in wheat plants at mock (H_2_O) treatment for 0.5 h is set to 1. (**B**) Transcript level of *TaAACT1* in leaves of wheat cultivar CI12633 after exogenous application of 0.1 mM ABA. The transcript level of *TaAACT1* in mock (H_2_O)-treated (0.5 h) wheat plants is set to 1. (**C**) Transcript levels of defense genes including *TaChit2*, *TaChit3*, *TaChit4*, and *TaDefensin* in wheat cultivar CI12633 leaves after exogenous application of 0.05 mM MeJA. The transcript levels of the tested genes in wheat plants at mock (H_2_O) treatment for 0.5 h is set to 1. Statistically significant differences (* *p* < 0.05, ** *p* < 0.01) are analyzed based on three replications using Student’ s *t*-test. Bars indicated the SD of the mean. *TaActin* was used as internal control.

**Figure 8 ijms-24-06165-f008:**
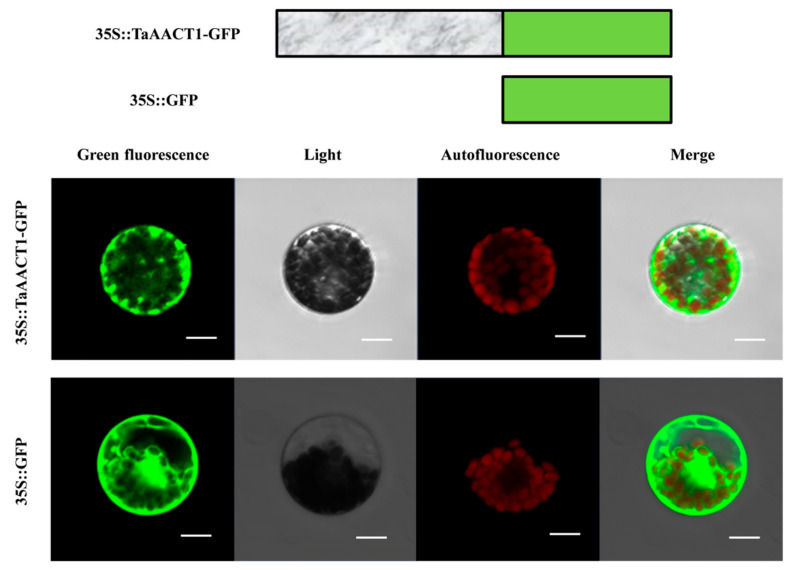
Subcellular localization of TaAACT1 in wheat mesophyll protoplasts. The fused TaAACT1-GFP protein and control GFP protein are transiently expressed in wheat mesophyll protoplasts. Scale bars = 20 μm. Green color represents the green fluorescence. Red color represents chloroplast.

**Table 1 ijms-24-06165-t001:** Nucleotide allelic variants of *TaAACT1*.

SNP Site (no.)	Nucleotide in Haplotype I	Nucleotide in Haplotype II	Amino Acid Change
2294	G	C	-
2360	T	C	-
2427	T	C	-
2449	T	A	-
2717	A	G	-
2938	G	A	-
3084	G	C	-
3114	G	A	-
3156	A	G	-
3264	T	C	-
3289	T	C	-
3304	T	C	S→P
3315	A	T	-

## Data Availability

All data supporting the findings of this study are available within the paper and within its Appendix A published online.

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
