# Peer review of "The Cytosolic Acetoacetyl-CoA Thiolase TaAACT1 Is Required for Defense against Fusarium pseudograminearum in Wheat"

_ijms, 2023, doi:10.3390/ijms24076165_

Round 1
Reviewer 1 Report
After carefully reading the title of the manuscript "“The cytosolic Acetoacetyl-CoA thiolase TaAACT1 is required 2 for defense against Fusarium pseudograminearum in wheat”, my comments are below:
- The manuscript is factual and appropriate for the International Journal of Molecular Sciences;
- The purpose of the work and the problems to be solved at work are clearly defined, the purpose of the work is within the scope of the IJMS;
- The results are interesting and important for researchers in the relevant fields of agricultural science as they relate to the health of one of the world's most important crops;
- The manuscript should be cited in the future and should be of great interest to readers in the research area represented.
I would like to propose some important corrections:
- line 4 - please remove: Authors;
- please read carefully the instructions for preparing the manuscript. In all components of the manuscript, the literature is incorrectly quoted, it should be changed in accordance with the requirements of the International Journal of Molecular Sciences!!! (order of cited items numerically in square brackets);
- Please increase the readability of all figures, especially 1-5;
- Line 333-334 - please correct the font size;
- The reviewed manuscript is properly prepared in terms of content,
Best regards,
Author Response
Dear Editors and reviewers
We are pleased tosubmit the revised manuscript. On behalf of all co-authors, we appreciate you and reviewers for the positive comments and constructive suggestions on our manuscript. We have thoroughly considered all the comments of the reviewers and substantially revised the manuscript. The responses to the reviewer’s comments are as follows.
Reviwer 1
Point 1: Line 4 - please remove: Authors.
Response 1: As requested, authors has been deleted.
Point 2: Please read carefully the instructions for preparing the manuscript. In all components of the manuscript, the literature is incorrectly quoted, it should be changed in accordance with the requirements of the International Journal of Molecular Sciences!!! (order of cited items numerically in square brackets).
Response 2: We have revised the format of all references in accordance with the requirements of the International Journal of Molecular Science.
Point 3: Please increase the readability of all figures, especially 1-5.
Response 3: We have revised the resolution of all the Figures in the article to improve the legibility of the Figures.
Point 4: Line 333-334 - please correct the font size.
Response 4: The font size and format have been modified as required.
Point 5: The reviewed manuscript is properly prepared in terms of content.
Response 5: Thank you very much for your approval of our experiment
We earnestly appreciate editors and reviewers for yourwork on our manuscript, and hope that the revisions are adequate.
We sincerely appreciate your time and efforts on our behalf and look forward to hearing your final decision.
With kind regards,
Yours sincerely,
Dr./Prof. Zengyan Zhang
Institute of Crop Sciences,
Chinese Academy of Agricultural Sciences,
- R. China

Reviewer 2 Report
The current manuscript identified a cytosolic AACT-encoding gene TaAACT1 and demonstrated its positively regulatory role in the wheat defense response to F. pseudograminearum. Firstly, they explore the allelic variant of TaAACT1 by analyzing the haplotypes of FCR-resistant and FCR-susceptible wheat cultivars and check the expression level of TaAACT1 respond to F. pseudograminearum infection in FCR-resistant wheat genotypes than in susceptible wheat genotypes. Then, they verified that the expression of TaAACT1 is required for resistance against F. pseudograminearum by VIGS. Also, they indicated TaAACT1 positively modulated the expression of downstream defense genes and TaAACT1 and its modulated defense genes were rapidly responsive to exogenous jasmonate treatment. This study investigates the function of a cytosolic AACT/thiolase II encoding gene in plant immunity and provides a potential gene for the molecular breeding of wheat with resistance to FCR. I advise an acceptance subject to revisions.
Line 78-89 Rewrite this paragraph. I am confused why the authors summarized the results and made a conclusion here.
Line 80 SNP should be haplotype or allelic variant.
Line 112 Restructure the table1 and Figure2 A, I hope the authors could indicate the polymorphisms in the gene structure.
The resolution of all the pictures must be improved.
Author Response
Dear Editors and reviewers
We are pleased tosubmit the revised manuscript. On behalf of all co-authors, we appreciate you and reviewers for the positive comments and constructive suggestions on our manuscript. We have thoroughly considered all the comments of the reviewers and substantially revised the manuscript. The responses to the reviewer’s comments are as follows.
Reviwer 1
Point 1: Line 4 - please remove: Authors.
Response 1: As requested, authors has been deleted.
Point 2: Please read carefully the instructions for preparing the manuscript. In all components of the manuscript, the literature is incorrectly quoted, it should be changed in accordance with the requirements of the International Journal of Molecular Sciences!!! (order of cited items numerically in square brackets).
Response 2: We have revised the format of all references in accordance with the requirements of the International Journal of Molecular Science.
Point 3: Please increase the readability of all figures, especially 1-5.
Response 3: We have revised the resolution of all the Figures in the article to improve the legibility of the Figures.
Point 4: Line 333-334 - please correct the font size.
Response 4: The font size and format have been modified as required.
Point 5: The reviewed manuscript is properly prepared in terms of content.
Response 5: Thank you very much for your approval of our experiment
We earnestly appreciate editors and reviewers for yourwork on our manuscript, and hope that the revisions are adequate.
We sincerely appreciate your time and efforts on our behalf and look forward to hearing your final decision.
With kind regards,
Yours sincerely,
Dr./Prof. Zengyan Zhang
Institute of Crop Sciences,
Chinese Academy of Agricultural Sciences,
P. R. China